# Engineering high quality graphene superlattices via ion milled ultra-thin etching masks

David Barcons Ruiz [1], Hanan Herzig Sheinfux[1], Rebecca Hoffmann[1], Iacopo Torre [1], Hitesh Agarwal [1], Roshan Krishna Kumar [1], Lorenzo Vistoli[1], Takashi Taniguchi [2,3], Kenji Watanabe [2,3], Adrian Bachtold[1,4] & Frank H. L. Koppens [1,4] ✉

Nanofabrication research pursues the miniaturization of patterned feature size. In the current state of the art, micron scale areas can be patterned with features down to ~30 nm pitch using electron beam lithography. Here, we demonstrate a nanofabrication technique which allows patterning periodic structures with a pitch down to 16 nm. It is based on focused ion beam milling of suspended membranes, with minimal proximity effects typical to standard electron beam lithography. The membranes are then transferred and used as hard etching masks. We benchmark our technique by electrostatically inducing a superlattice potential in graphene and observe bandstructure modification in electronic transport. Our technique opens the path towards the realization of very short period superlattices in 2D materials, but with the ability to control lattice symmetries and strength. This can pave the way for a versatile solid-state quantum simulator platform and the study of correlated electron phases.

Nanoscale fabrication is at the heart of the technological revolution of semiconductor technology and scientific breakthroughs in nanotechnology, progressing rapidly for more than five decades[1]. Accordingly, the push to further improve nanofabrication techniques is an ongoing effort to fit more electronic components per unit area or to develop quantum devices and quantum bits, relying on quantum coherent control in the nanoscale regime[2,3]. Another important application is in quantum materials, where materials' properties can be tuned in situ. Arrays of quantum dots with sufficiently large Coulomb interactions could lead to the observation of metal-Mott insulator quantum phase transitions and potentially even the d-wave superconducting phase[4]. Recent discoveries of correlated phases in twisted bilayer graphene due to the superlattice (SL) potential[5,6] motivate further exploration of graphene SLs with more versatility in the lattice design and in situ control.

Nanofabrication techniques always involve a trade-off between patterning resolution, throughput and flexibility. For example, scanning tunneling microscopy, can be utilized to rearrange materials on the atomic scale[7,8], but is impractically slow for micron-scale lithography. Other techniques, such using self-assembled polymers, can achieve high resolution and yield[9], at the expense of limiting design flexibility in what patterns can be implemented. A higher throughput alternative is ultraviolet optical lithography, the main lithography technique in the semiconductor industry, which is faster but is not suited for academic research and development.

In academic research, the most common nanopatterning technique is based on electron beam lithography (EBL), which can be thought of as a compromise offering reasonable throughput and nanoscale resolution. Forward scattered electrons limit the minimal patternable feature size, while secondary electrons due to (inelastic scattering)

[1]ICFO—Institut de Ciencies Fotoniques, The Barcelona Institute of Science and Technology, 08860 Castelldefels (Barcelona), Spain. [2]Research Center for Functional Materials, National Institute for Materials Science, Tsukuba, Japan. [3]International Center for Materials Nanoarchitectonics, National Institute for Materials Science, Tsukuba, Japan. [4]ICREA—Institució Catalana de Recerca i Estudis Avançats, 08010 Barcelona, Spain. ✉e-mail: frank.koppens@icfo.eu

interactions with the resist limit the pitch resolution. As such, using high acceleration voltages and thin organic resists, patterns with 35 nm pitch can be obtained, but significant reduction of the feature size has not been demonstrated[10–12].

Another common nanofabrication approach uses focused-ion beam (FIB) milling to pattern the target material directly. In particular, helium (He) FIB routinely achieves single digit single feature resolution and pitch on the order of 10 nm[13–18]. However, two major limiting factors affect He-FIB-based nanofabrication: (1) Milling of materials directly on a substrate is accompanied by deposition of He bubbles inside the substrate, contaminating it and affecting surface topography[18,19]. (2) Secondary collisions damage the quality and integrity of the patterned material[19,20]. It is possible to reduce the damage by milling suspended materials (where substrate scattering is absent) or using resist lithography[17,21]. However, suspending the material is complex and the amount of FIB induced unintentional damage is still expected to modify the material properties in the form of ion implantation and/or amorphization[20,22–24]. Here, we propose and demonstrate a nanofabrication technique that combines the high resolution of He-FIB milling with minimal damage to the patterned material.

Our technique uses ultrathin suspended hard masks that are placed in contact with the target substrate material. This alleviates proximity effects and feature broadening induced by scattering from the substrate. The suspended ultrathin mask can be subjected to harsh FIB patterning with minimal proximity effects and later transferred onto the target material to project the pattern by reactive ion etching (RIE). The resolution of our method is demonstrated directly by atomic force microscopy (AFM), whereas the quality is demonstrated by applying our technique to engineer SL potentials in graphene via patterned graphite electrodes. Graphene could be very sensitive to the presence of charge traps, impurities and patterning errors in the patterned gate electrode. However, transport data from our samples indicates minor unintentional damage to the patterned electrodes, even for highly dense patterns with an 18 nm pitch. Similar to moiré systems (e.g. twisted graphene[25] or graphene/hexagonal boron nitride (hBN) heterostructures[26,27]), it is possible to engineer SLs by applying a periodic electrostatic potential[28]. When applied to low-dimensional materials, such as graphene, this modifies its electronic band structure when the SL period is much smaller than the electronic coherence length[29]. Several approaches have been explored to introduce such potentials: from patterning the graphene directly[12], to inducing a periodic potential modulation by patterning the dielectric material[30,31] or the gate electrode[32–34]. In all cases, it is essential to reduce damage to the patterned material, keeping it flat and free of external contamination. However, to realize the full potential of artificial SLs, a new lithographic technique is required, one which can create high quality periodic patterning on the sub-20-nm scale, approaching the moiré length in graphene aligned on hexagonal boron nitride or magic angle twisted bilayer grapheme (Fig. 1).

## Results and discussion
### Ion-milled hard mask lithography
As the hard mask, we use a poly-crystalline silicon suspended ultrathin membrane (5–10 nm), which is commercially available on a large scale and high quality. Importantly, the suspended membrane can be removed from its supporting frame and transferred with a polymeric stamp (Fig. 2b). The membrane is placed on top of a mechanically exfoliated target graphite flake, coated by a thin 22 nm layer of poly(methyl methacrylate) (PMMA) (Fig. 2c), which serves as a sacrificial layer for later removal of the membrane (we note it is possible to remove the membrane in other ways, namely etching, but we found the use of PMMA layer to be more robust, see Supplementary Note 3). Due to the nanometric thickness of the hard mask, the aspect ratio is not a limitation and the feature size is only limited by the ion beam milling process. After the Ar/O$_2$ RIE process, we remove the hard mask (Fig. 2d) with a standard acetone lift-off followed by vacuum annealing to remove polymer surface contamination. As a demonstration of the capabilities of our technique, we achieve a triangular lattice with a period as small as 16 nm and a hole diameter down to ∼8 nm (Fig. 2e).

To quantify the disorder of our lattices, we perform Fourier analysis of the AFM images (Fig. 2f), finding less than 3% period variations in our smallest lattices (16 nm period). By inspecting the transmission electron microscopy (TEM) images of our He-FIB milled silicon masks, we find a diameter of the holes of $13.2 \pm 0.4$ nm for a 22 nm period lattice, and thus the variation in diameter is only 2% and less than 6% in area. To showcase the high quality achieved with our technique, we incorporate a patterned graphite gate into two different SL graphene devices. Both are based on encapsulated single layer graphene with a square periodic lattice pattern in the bottom gate electrode (Fig. 3a), but with different period—$a_{SL} = 47$ nm (Dev 1) and $a_{SL} = 18$ nm (Dev 2), and different hBN spacer thickness—$t_{BN} = 6.2$ nm (Dev 1) and $t_{BN} = 3.2$ nm (Dev 2).

### Electronic transport measurements
Figure 3c shows a zoom-in of the AFM characterization of the patterned gate electrode of Dev 1. The square SL in single layer graphene is expected to lead to the emergence of cloned Dirac cones equally spaced in energy due to band folding in the mini-Brillouin zone[32]. By tuning the silicon backgate voltage (Si BG) and the patterned gate voltage (PBG), one can modulate the strength of the SL and the carrier type in the graphene layer to observe the cloned Dirac cones, see Fig. 3e. When the Si BG is kept at 70 V, two satellite peaks (indicating cloned Dirac cones) for electrons and another two for holes are clearly visible, being roughly of the main size as the main Dirac peak. This contrasts the situation where the Si BG is kept at 0 V, where we do expect a very small density modulation and indeed no satellite peaks are observed. The carrier density is normalized by the number of electrons per SL unit cell, $n/n_0$, and therefore, the spacing of the satellite indicates the four-fold (spin and valley) degeneracy in graphene. The clear observation of multiple satellite peaks, and their similar width and prominence compared to the main Dirac peak,

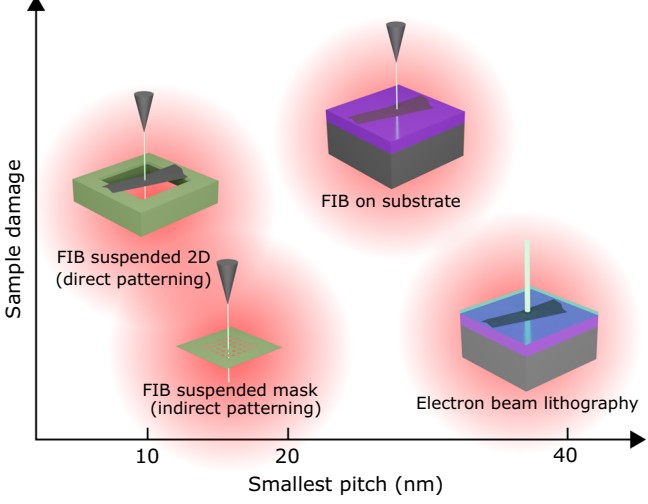

**Fig. 1 | Comparison chart of focused-ion beam (FIB) alternatives to standard electron beam lithography (EBL) for 2D materials patterning.** While FIB improves the resolution limit of EBL, substrate damage has to be considered. Suspending the 2D material for the patterning process alleviates substrate swelling compared to standard silicon wafer substrates but introduces disorder in the crystalline lattice. As an alternative, indirect FIB patterning of a suspended membrane, later used as an etching mask for the 2D material, results in no substrate damage and still preserves the high resolution of suspended FIB patterning.

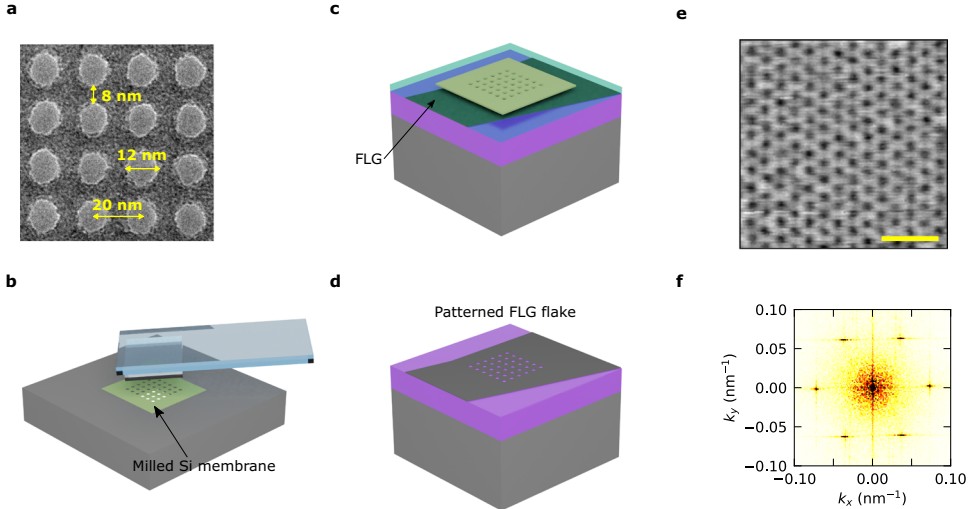

**Fig. 2 | Fabrication process of a patterned few layer graphene (FLG) gate electrode. a** Transmission electron microscope (TEM) image of a thin suspended silicon membrane which has previously been milled with a He focused-ion beam (FIB). **b** The membrane is transferred with a polypropylene carbonate (PPC) coated polydimethylsiloxane (PDMS) stamp onto a FLG flake coated with a thin layer of poly(methyl methacrylate) (PMMA). **c** The sample is etched following a standard O₂/Ar reactive ion-etching (RIE) process, followed by a lift-off process to remove the membrane and clean the PMMA layer underneath (**d**). **e** Atomic force microscopy (AFM) topography image of a 16 nm period triangular lattice on a FLG flake. The scale bar is 50 nm. **f** Fast Fourier transform (FFT) of a larger region of the same AFM image in panel **e**, including 2555 lattice sites.

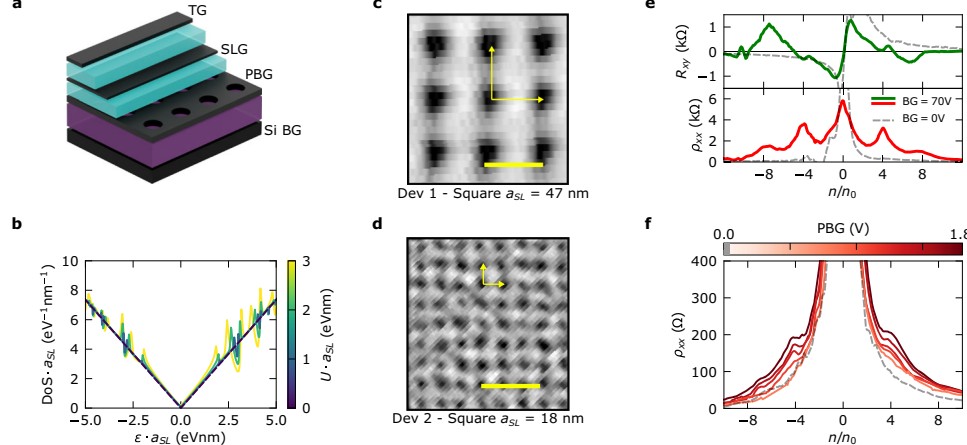

**Fig. 3 | Electronic transport characterization of devices Dev 1 and Dev 2. a** Heterostructure schematic. The main difference between Dev 1 and Dev 2 is the use of a graphite top gate in the latter. **b** Predicted normalized density of states per superlattice (SL) length $a_{SL}$, which demonstrates the invariance of the Dirac equation with the product $a_{SL}$ and the potential modulation $U \cdot a_{SL}$. **c, d** Atomic force microscopy (AFM) topography images of the patterned graphite gates used for each device. The scale bars are 50 nm. **e** Longitudinal resistivity (red trace) and Hall resistance at $B = 0.1$ T (green trace) as a function of the electron density per SL unit cell for Dev 1 (silicon backgate voltage $V_{BG} = 70$ V). Gray dashed traces display longitudinal resistivity and Hall resistance for $V_{BG} = 0$ V, i.e., when there is no SL modulation. In the first situation, clear satellite peaks can be observed when the normalized density $n/n_0$ is a multiple of 4, with $n_0$ corresponding to the density where each SL unit cell is filled with one electron. This is consistent with the graphene's four-fold degeneracy. **f** Longitudinal resistivity for Dev 2 as a function of the patterned bottom gate (PBG) applied. Satellite peaks emerge as well at normalized densities $n/n_0 = \pm 4$. Gray dashed line indicates $V_{PBG} = 0$ V, for which no satellite peak is observed.

demonstrate the high quality of the device and the high uniformity of the patterned gate.

The second device studied in this work, Dev 2, has an 18 nm period square SL. To the best of our knowledge, there is no experimental work about electrostatically induced SLs with such a short periodicity. In this case, there is a top gate electrode (TG) apart from the PBG. This combination of gates allows us to control the SL strength and the total carrier density of the system independently. Therefore, we keep PBG fixed at certain voltages and sweep TG to change the carrier density. In longitudinal resistivity, we observe clear signatures of the emergence of satellite peaks at $n/n_0 = \pm 4$, as PBG is increased (Fig. 3f).

Furthermore, when the PBG is set at 0 V (gray dashed line in Fig. 3f), i.e., no carrier density modulation, we do not observe any superlattice feature. The observed dependence of the SL features on the PBG confirms its electrostatic origin. We cannot completely discard the possibility there is unintentional double alignment to the hBN flakes. This would be an additional effect on top of the electrostatically induced SL. We discuss this possibility and show additional data in Supplementary Note 3 and 7.

The prominence of the SL signature is different in the two devices. To understand this, we must examine the specific characteristics of both devices. As a rough approximation for the electrostatic potential

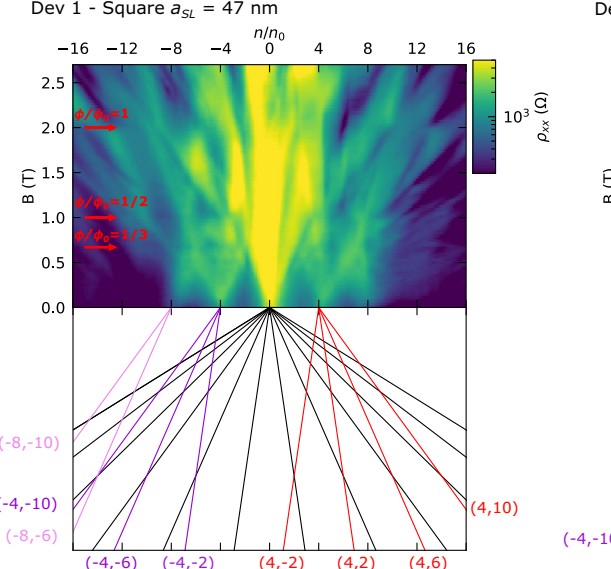

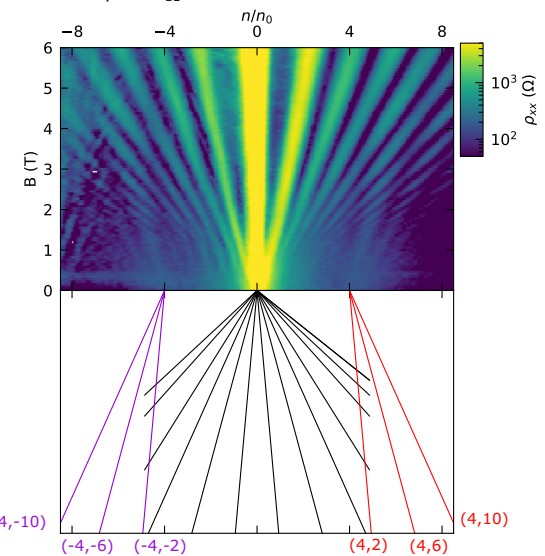

**Fig. 4 | Longitudinal magneto-resistance measurements of devices Dev 1 and Dev 2.** For Dev 1 ($V_{BG} = 70$ V), we observe a rich Hofstadter butterfly where Landau fans emerge from filling fractions $n/n_0 = 0, \pm 4, \pm 8$. Red arrows indicate resistance dips corresponding to an integer fraction of the flux quantum $\phi/\phi_0 = 1/3, 1/2, 1$. For

Dev 2 ($V_{PBG} = 1.4$ V), Landau fans emerge from $n/n_0 = 0, \pm 4$. These are solid evidence for the formation of an 18 nm period superlattice (SL). Colored lines below the magneto-resistance maps indicate the observed Landau fans. Labels indicate the SL filling fraction and Landau level ($n/n_0$, LL).

modulation along the graphene layer for the formation of the SL, we consider a periodic one-dimensional system with a patterned bottom gate, global top gate (more details in Supplementary Note 4), and 50% duty cycle gate/vacuum. Due to the linearity of the Poisson equation, the induced modulation of the electrostatic potential $U$ scales with the ratio of the dielectric thickness $t_{BN}$ and the SL period $a_{SL}$. We estimate for Dev 1 ($t_{BN}/a_{SL} = 0.13$) $U \sim 200$ meV, while for Dev 2 ($t_{BN}/a_{SL} = 0.18$) $U \sim 120$ meV. Furthermore, we need to consider the scaling of the Dirac equation (Eq. 1). The larger the external potential term $U(x,y)$ is compared to the unperturbed graphene Hamiltonian $\mathcal{H}_0$, the stronger SL effects in the band structure will be:

$$\mathcal{H} = \mathcal{H}_0 + U(x,y)\mathbb{I} \tag{1}$$

Since the momentum at the mini-Brillouin zone edge is inversely proportional to the SL period $a_{SL}$, decreasing $a_{SL}$ by a factor $\alpha$ enhances the unperturbed graphene term compared to the electrostatic potential term. In order to compensate for this enhancement, the external potential $U(x,y)$ would need to be larger by the factor $\alpha$. At the same time, the energy of the system will scale inversely proportional to $\alpha$. As a result, the comparatively small $U$ and small $a_{SL}$ in Dev 2 makes that the effect of the electrostatic modulation and, thus, the band structure modification in Dev 2 is weaker than that in Dev 1. Figure 3b displays the importance of the Dirac invariant parameter $U \cdot a_{SL}$ ($\sim 9.4$ for Dev 1 and $\sim 2.2$ for Dev 2) on the density of states.

**Magneto-transport measurements**

Further evidence of the modified band structure is given in Fig. 4, where the gate voltage configurations are the same as in Fig. 3. For Dev 1 ($a_{SL} = 47$ nm), Landau fans emerge from the satellite peaks at filling fractions $n/n_0 = 0, \pm 4, \pm 8$. Given the relatively large period compared to typical moiré systems, it is possible to reach one quantum of magnetic flux inside the SL unit cell. We also observe resistance dips at integer fractions of the flux quanta $\phi/\phi_0 = 1/3, 1/2, 1$, indicated by red arrows. They could be attributed Brown-Zak oscillations[35,36], but further investigation would be needed in order to confirm this. For Dev 2 ($a_{SL} = 18$ nm), apart from the LL spectra emerging from charge

neutrality, two other fans converge to densities $n/n_0 = \pm 4$. This is solid evidence for our device's 18 nm period SL formation. Understanding the origin of additional oscillations between $n/n_0 = 0$ and $n/n_0 = -4$ at finite magnetic field is subject of future work.

## Discussion

To conclude, we have successfully demonstrated a nanofabrication technique that strongly alleviates proximity effects allowing to go beyond the spatial resolution limits of EBL. We used this technique to induce a SL potential in single layer graphene devices. To demonstrate the formation of the SL, we have presented electron transport data showing satellite peaks corresponding to cloned Dirac cones, Hofstadter butterfly spectrum, and Landau levels emerging from the satellite peaks. To the best of our knowledge, the dimension of our smaller SL with $a_{SL} = 18$ nm sets a record, enhancing by a factor of four the relevant Coulomb interaction strength $\propto 1/a_{SL}^2$ [12,30–32].

However, there are limitations in the nanopatterning process that remain to be improved in future works. In particular, it remains challenging to pattern long lines or closed shapes, such as rings, due to the suspended nature of the masks. Furthermore, the use of a PMMA sacrificial layer and the substrate back-scattered ions during the RIE could be sources of damage and contamination in our samples, despite the cleaning efforts. Possible routes, such as adding an hBN flake at the bottom of the patterned gate[12], or integrate the silicon mask directly in the heterostructure, could improve the process quality.

The ability to engineer arbitrary lattice geometries opens the path toward studying non-bipartite lattices[37] and flat bands in Dirac and gapped Dirac systems, such as the Lieb or Kagomé lattices, which require superior spatial resolution compared to patterned SLs achieved thus far. Furthermore, our technique enables a new generation of Fermi-Hubbard model simulators[38] when combining our patterned gate electrodes with 2D tunable semiconductors such as transitions metal dichalcogenides[39] or bilayer graphene[40,41]. Combining the patterned gate with a second gate allows the carrier filling and the Hubbard on-site interaction strength $U$ to be tuned independently. The superior quality of our nanopatterning process will make it possible to engineer lattices where the Hubbard $U$ can reach the

10–100 meV range, such that exotic correlated phenomena can be engineered at comparatively high temperatures.

## Methods

### Hard mask FIB milling

Fabrication of the devices begins with commercially available poly-crystalline silicon membranes (US100-A05Q00, SiMPore). Prior to milling in FIB, the membranes are annealed in a $H_2$:Ar atmosphere, at 400 C, for 3 h. We find that this step of annealing greatly reduces mechanical strain in the Si membranes. On inspection in optical microscope, the membranes appear slightly wrinkled before the annealing process, whereas after annealing the membranes noticeably flatten.

The membranes are mounted on a hole in the sample holder, to improve imaging contrast and loaded into the He focused-ion beam microscope (HIM). The HIM is then aligned to high precision on metallic features in the sample, with typical parameters being 5 μT He pressure, 10 or 20 μm aperture and spot control between 3 and 5. This yields a beam current between 2 and 12 pA and resolution between $\sim$2 and 10 nm (determined by visualizing sharp edges and features), with the exact milling parameters determined to match the achievable resolution with the desired periodicity of the pattern. Notably, the use of relatively high currents allows faster milling and reduces mechanical drift and similar undesirable effects.

After initial alignment, the membrane area is located in the HIM (the thin membrane shows clear contrast with the thicker, more conductive, background surrounding it). The sample is usually allowed to rest for 30–90 min (overnight when time allows), to minimize the effects of piezo drift in the sample stage. Following this resting period and additional finer alignment, the membrane is milled using a pre-programmed point-by-point deflection list. For large features, a milling dose comparable to 0.9 nCμm$^{-2}$ is used. For smaller features, and especially when the lattice period approaches the HIM resolution (for the specific parameters used), the milling dose is optimized via demo patterns, where milling is typically sequential (single repeat rather than multiple repeats). The size of the milled area is typically kept on the order of $3 \times 3$ μm$^2$ in accordance with experimental needs. After milling the desired patterns, 20 nm wide cuts are made to the side of the membrane, leaving it partially connected to the frame via 3–4 μm wide connecting bridges. These cuts facilitate breaking the membrane later during the transfer process.

In addition to the majority of the samples produced during our research, using He-FIB, we were also able to produce similar results (but with larger milling periods) using a Gallium focused-ion beam microscope (Ga FIB). These samples were processed similarly to the above described samples milled by HIM, with the exception being that the membrane was mounted with the window facing down (situated above a hole in the sample holder), in order to facilitate locating the intended milling area with the FIB. The resolution of the specific Ga FIB microscope used in our experiments was restricted by various technical issues, limiting the size of the spot effected by the Ga ions to about 20 nm and the lattice period to the order of 50 nm. With a more narrowly focused FIB, we estimate, based on our experience, that a ~10 nm single feature and ~30 nm lattice period should be achievable.

### Device fabrication

The fabrication process of the patterned graphite gate is explained in the main text. After the cleaning process, we prepared an hBN/graphene/hBN (hBN/graphene/hBN/graphite/hBN in the case of Dev 2) heterostructure. We first dropped it on a clean Si/SiO$_2$ substrate at a temperature of 158 °C to clean its interfaces, and subsequently temperature was increased to 180 °C to melt the poly (bisphenol A carbonate) (PC) film[42]. After dissolving the PC film in chloroform, the heterostructure quality is checked through AFM imaging and Raman spectroscopy. Finally, the heterostructure is again picked up and released following the same procedure on the patterned graphite gate. It is important to note that the hBN flakes between the graphene layer and the patterned gate electrode must be thin to allow for efficient doping modulation.

Standard EBL is used to pattern the Hall bar geometry and make one-dimensional edge contacts[43] to our heterostructure. In the case of the contacts, since they are right on top of the patterned graphite flake, we perform first an SF$_6$ etching process[44] to remove only the top hBN, followed by an O$_2$/Ar etching to remove the graphene, but not the thin bottom hBN that will prevent leakage current between the bottom graphite gate and the electrodes. We deposit Cr(3 nm)/Au(40 nm) followed by a lift-off in acetone.

### Electronic transport measurements

Electrical measurements were performed in a He flow cryostat from ICE Oxford operating at $T = 1.45$ K. Measurements were taken using standard lock-in techniques. A constant current of 5–20 nA was sourced by using a 10 MOhm resistor in series with our device at frequencies 11–18 Hz. Si BG, PBG, TG were controlled independently with a source meter.

### AFM images processing

To remove substrate height variations, we apply a high pass filter with a cut off frequency $f = 1/3 \cdot f_{a_{SL}}$ by performing a 2D fast Fourier transform. See Supplementary Note 8 for an example of the pre-processed and post-processed images.

## Data availability

Relevant data supporting the key findings of this study are available within the article and the Supplementary Information file. All raw data generated during the current study are available from the corresponding author upon request.

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

## Acknowledgements

D.B.R., H.H.S., R.H., I.T., H.A., R.K.K., L.V., A.B., and F.H.L.K. acknowledge funding from the Government of Spain through CEX2019-000910-S [MCIN/AEI/10.13039/501100011033], Fundació Cellex, Fundació Mir-Puig, and Generalitat de Catalunya through CERCA. D.B.R. acknowledges funding from the Secretaria d'Universitats i Recerca del Departament d'Empresa i Coneixement de la Generalitat de Catalunya, as well as the European Social Fund (L'FSE inverteix en el teu futur)-FEDER. H.H.S. acknowledges funding from the European Union's Horizon 2020 programme under the Marie Skłodowska-Curie grant agreement Ref. 843830. H.A. acknowledges funding from the European Union's Horizon 2020 research and innovation program under the Marie Skłodowska-Curie grant agreement no. 665884. R.K.K. acknowledges the EU Horizon 2020 program under the Marie Skłodowska-Curie grants 754510 and 893030. L.V. acknowledges funding from the H2020-MSCA-IF-2019 [887367-NanoMagnO]. A.B. acknowledges support from ERC Advanced Grant No. 692876 and MICINN Grant No. RTI2018-097953-B-I00, the European Union's Horizon 2020 research and innovation programme under the Marie Skłodowska-Curie grant agreement No. 847517 and 101023289, AGAUR (Grant No. 2017SGR1664), the Quantera grant (PCI2022-132951), the Fondo Europeo de Desarrollo, Recovery, Transformation and Resilience Plan-Funded by the European Union-NextGenerationEU and Quantum CCAA. F.H.L.K. acknowledges support by the ERC TOPONANOP (726001), the Government of Spain (PID2019-106875GB-I00), and Generalitat de Catalunya (AGAUR, SGR 1656). Furthermore, the research leading to these results has received funding from the European Union's Horizon 2020 under grant agreement no. 881603 (Graphene flagship Core 3) and 820378 (Quantum flagship).

## Author contributions

All authors contributed to writing the manuscript. H.H.S. fabricated the silicon masks, and D.B.R. and R.H. fabricated the devices, with help from H.A. Measurements were taken by D.B.R. with help from R.K.K., L.V., and H.H.S. D.B.R. and H.H.S. performed the data analysis. T.T. and K.W. provided the hBN crystals. I.T. performed the electrostatics and band structure calculations. F.H.L.K. and A.B. supervised the work.

## Competing interests

The authors declare no competing interests.
