## [Peer Review File · Nature Communications]

Reviewers' Comments:

Reviewer #1:

Remarks to the Author:

D. B. Ruiz and coworkers present work on artificial electrostatic superlattices in 2D materials. The key novelty of the work is to use a focused ion beam pre-patterned membrane to transfer a pattern into a graphite gate, which in turn can be used to electrostatically induce a superlattice in a 2D material. This is shown convincingly for a 47 nm pitch device, which is somewhat larger than the state-of-the-art for other methods. The main selling point of the paper, however, is an 18 nm pitch device. Unfortunately, the authors do not demonstrate a gate-gate map for this device, and several features of the magnetotransport are simply ignored. Mostly, the device resembles a normal graphene device with single or double alignment of the encapsulating hBN layers. The paper contains an interesting discussion of how the strength of the superlattice scales with pitch, but it is ultimately somewhat decoupled from the rest of the paper until convincing measurements can be shown for a <20 nm pitch device.

The method presented has the potential to be of interest to the 2D superlattice community. However, the data, as presented in the manuscript, does not support the claims of the paper, namely that superlattices in the 16-18 nm range can be fabricated and integrated into a device structure.

As such, I do not believe the paper is suitable for Nature Communications.

Below I give a list of specific comments.

Specific comments and questions:

9 of the 35 references are either review papers or textbooks, raising questions about how well the authors researched the literature.

Abstract

"with minimal proximity effects typical to electron beam lithography"

Suspended membranes also alleviate proximity effects in EBL.

"shows high quality superlattice properties"

Figure 4A seems of high quality, but similar to previously reported devices.

It is not clear from device 2 that the superlattice has any effect

Introduction

In general, the introduction is littered with statements that are not backed up with any references.

"extreme ultraviolet optical lithography (EUV), the main lithography technique in the semiconductor industry."

DUV and EUV are not the same. EUV is used for only very few layers of the leading edge nodes at TSMC. Not even Intel is shipping products using EUV yet.

"This type of nanofabrication can achieve small feature sizes^{7,8} and is highly scalable, but is prohibitively complicated and expensive."

What is the context here? EUV is expensive, indeed, but not prohibitively so for industry. The introduction is quite messy, and mixes up the narrative between industry applications and academic prototyping needs.

"However, two major limiting factors affect He-FIB-based nanofabrication: (1) Milling of materials directly on a substrate is accompanied by deposition of He bubbles inside the substrate,

contaminating it and affecting surface topography. (2) Secondary collisions damage the quality and integrity of the patterned material. It is possible to reduce the damage by milling suspended materials"

At this point, only two papers are referenced about FIB lithography. Both extensively discuss suspended samples and membranes, overcoming (1) and (2). Why are these issues brought up?

FIB indirect patterning of 2D materials

"The membrane is placed on top of a mechanically exfoliated target graphite flake, coated by a thin 22 nm layer of poly(methyl methacrylate) (PMMA)"

Could you elaborate on the role of adding PMMA? I assume this serves as a sacrificial release layer?

The patterned membrane is characterized by TEM, while the patterned PMMA/graphite stack is characterized by AFM. While AFM, through the FFTs, can to some degree give information about the geometrical disorder, it would be beneficial to do TEM imaging of graphite after pattern transfer, to measure the edge roughness and atomic disorder, as these are some of the main claims of the paper.

Also, given how pixelated the FFTs are, especially the inset, I don't think it is appropriate to use these for quantitative extraction of disorder. Zero padding the data prior to the transform might help here.

Electronic transport characterization

"This combination of gates allows us to control the SL strength and the total carrier density of the system independently "

Why is this not shown? To prove the effect of the superlattice you should provide a gate-gate map. This is especially critical for Dev.2. where I am not convinced about the quality of the device, and whether or not it is simply aligned to one of the hBN layers.

Magnetotransport of device 1 looks convincing, especially with the gate-gate map in the supporting information, but the length scale has already been reported in the literature previously. The quality of the 47 nm device suggests that the presented method could give better device quality compared to the previously reported methods based on electron-beam lithography, but to support this, the authors would have to do a more careful analysis of both the transport data and the quality of the fabricated membranes and pattern transfers.

"For Dev 2 ($a_{SL} = 18$ nm), apart from the LL spectra emerging from charge neutrality, two other fans converge to densities $n/n_0 = \pm 4$. This is solid evidence for our device's 18 nm period SL formation."

This is absolutely not the case. Without gate-gate maps and additional characterization, there is no reason to believe that the features observed are not due to alignment with one or two of the encapsulating hBN layers. Furthermore, there are several features in the fan diagram that are completely ignored, e.g. several signatures of satellite fans between $n/n_0 = 0$ and 4.

Reviewer #2:

Remarks to the Author:

In their manuscript "Engineering high quality graphene superlattices via ion milled ultra-thin etching masks", Ruiz et al present a fabrication method for extremely finely pitched 2D superlattices by using Helium (or Ga) FIB. By first milling an ultrathin membrane and only then transferring it to the target substrate, milling and removing the template, they circumvent the problem of extensive contamination or milling damage usually found in FIB based lithography. With their approach they can extend the range of achievable feature resolution from a ~ 35 nm lattice period, achieved in state-of-the-art e-beam lithography, down to about ~ 18 nm. They integrate their superlattices into a high quality graphene workflow and demonstrate successful definition of a tunable superlattice potential in graphene.

The fabrication and experimental results are astounding, and present a significant step forward in

nanofabrication, at the expense of very specialised equipment (He-FIB) and a quite involved workflow. The results definitely merit publication, but I would like to suggest a few improvements, mainly regarding the presentation of the results.

* In the Supplemental Material, the authors quite openly discuss a few issues with the presented technique (section 3, lines 80..) A summary of those (and a few more) points should make it into the conclusion of the main text, otherwise the conclusion sounds too optimistic. In addition to what is mentioned in the Supplemental Material, I would like to add that a 2D hole array is the most stable pattern imaginable. I think that fabricating, e.g., a stripe array with a similar pitch, would pose extreme challenges in terms of stability, and a non-connected geometry is evidently impossible to make with a stencil technique. The authors should address those issues in the main text, even though the new technique will still remain extremely useful for quite a few applications.

* Can Si membranes be etched in the present workflow without damaging the substrate? This could remove the need for a PMMA based lift-off. Maybe some F-based dry etching could work?

* Brown Zak oscillations (Fig 4, and lines 152...) are more easily observed at elevated temperatures, when Landau fans are already washed out. One can argue that BZ oscillations are just the remnants of the Hofstadter butterfly skeleton (which is what the authors actually see in Fig 4), but at least the authors of Ref 27 would probably disagree with this. Maybe some magnetic field sweeps at around 100 K and fixed density are available, or can be supplied?

* The authors briefly mention that they also tried Ga FIB (which is available in more labs than He FIB). Can they be more specific as to the minimum feature size realized here, and supply some images? The present statement in SM, l. 52 is too vague.

* Can the authors provide the thicknesses of hBN separating patterned gate and graphene channel? They can be inferred from l. 132 and l. 133 but the authors could just as well provide the values. Also, real images of the actual devices should be supplied, at least in the Supplemental Material.

Reviewer #3:

Remarks to the Author:

In this paper, a new method of fabricating superlattice devices is reported, in which a suspended Si membrane is etched by ion milling to create an etching mask. This mask is transferred onto a thin graphite flake so that it can be used to etch a superlattice gating structure. Using this technique, the authors are able to create a lithographically defined superlattice device with significantly shorter periodicity than other methods with similarly low disorder, namely methods relying on e-beam lithography to define the mask.

The measured device resistance (Figure 4) indicates a Hofstadter butterfly spectrum as is expected in these devices. Given the short periodicity in device 2 (18nm), which increases the spacing of fractal band features in both electron density and magnetic field, it is rather disappointing that the Hofstadter effect is so weak. I would hope that the smaller periodicity would give rise to a more brilliant butterfly; that this increased spacing would allow for greater resolution of fractal subbands. The authors give a well reasoned explanation for why this effect weakens as the periodicity is shortened. This perhaps elucidates an inherent weakness of using electrostatically induced SLs as a means of replicating the effects of small-angle moire superlattices.

Regardless, this fabrication technique is a significant step forward for patterning gating structures with nanometer scale features. There is a wide audience in the layered heterostructure devices community who would be interested in this result and potentially find use for this technique, especially since it can be applied to more than superlattices. Therefore, I support its publication in Nature Communications.

Comments:

39- Opening sentence states a trade-off, resolution vs throughput. EUV is given as an example of the low-resolution, high-throughput end of the spectrum, but it's described as having "small

feature sizes" but complicated and expensive. Perhaps give actual numbers for the features sizes STM vs EUV to elucidate this comparison?

More generally, I found this discussion of resolution vs throughput somewhat irrelevant to the academic setting where EBL is very common. I never considered using STM for making a superlattice (should I?)

71- Specifically, how does the transport data indicate minor damage to the electrodes? Can you measure the mobility in your devices (or mean free path)?

From your transport data, it is clear that there is no breaking of graphene's 4-fold degeneracy, which sets these devices apart from moire superlattice devices. Of course, this is true of all previous attempts to create a lithographically defined superlattice as well. My point is, there is still a huge gulf between a normal encapsulated graphene device (with top and bottom graphite gates, no less) and devices like these, in terms of mobility, degeneracy breaking, and true quantization of Hall resistivity. Pointing to specific signifiers of device quality is valuable here.

90- Can you just state the thickness of the hard mask here? I think it's helpful context for the reader. Looking at the supplementary, it's of order 10 nm?

119- "On the other hand, the PBG only changes the carrier density on the areas of graphene right on top of the remaining graphite, while the areas on top of the patterned holes remain unaffected" This isn't really true though, right? Looking at your simulation results (Figure S3), you can see that the carrier density within the holes is affected by the graphite gate voltage as well. I think if your model was 3D with an actual circular hole, I think you'd see an even larger effect within the hole region.

It's fair to say that you can independently tune the carrier density and SL strength, but the effect of the PBG is more complicated than it only affecting the graphene above the unetched regions. This thickness, t_{BN} , is also critical here for how much the hole regions are affected by PBG voltage.

Fig 1, line 70, line 80 - This paper's characterization of disorder in these samples is lacking any mention of the disorder created by RIE etching of the PBG or exposing it to PMMA. I am not sure where in the paper this could be added, but I think a recognition of this source of disorder would be valuable. The text mentions the importance of considering charge traps (line 70) and external contamination (line 80), both of which I think are potentially created through the PMMA and RIE etch. Though I recognize that annealing the PBG following the acetone liftoff is fairly effective, it is still worse than no PMMA contamination. Also, this sort of contamination is shared by the EBL defined techniques which are referenced in this paper.

To be clear, this technique is still very valuable, as it allows FIB to be used in devices with similar quality to those produced by EBL; so Figure 1 is still valid.

Fig. 3a: This attempt to show the device, using transparent layers, is not really legible. I don't think it will be helpful for anyone who doesn't already know how these devices are stacked. Perhaps try removing metal leads and the "Hall bar" shape of the device, and just focus on the layers themselves.

Fig 4a- The red ϕ/ϕ_0 labels are unreadable. Make them bigger? Bold font? Make them another color (white?)?

I like the Conclusion of the paper. I think that looking at new lattice geometries and Fermi-Hubbard model simulators are promising next steps for this sort of FIB mask transfer technique. I also think that this paper shows a difficult hurdle for engineering Hofstadter butterflies: that the lattice strength at the cloned Dirac bands weakens as the SL period is shortened. This is all the more reason to shift attention to new applications for this technique.

General comments

Our paper has been reviewed by three referees. All three referees agreed our results are of interest to the scientific community, so in that regard should be suitable for publication in nature communications. However, while referee #2 and #3 were supportive of publishing our paper, referee #1 raised important concerns about the possibility our results stem from alignment to hBN, rather than electrostatic modulation. In our detailed rebuttal below, we will conclusively demonstrate this cannot be the case.

In addition, all three referees provided very valuable comments and suggestions in their very detailed referee reports. We are very thankful to all three for their effort.

Reviewer #1 comments

Reviewer #1 states “The method presented has the potential to be of interest to the 2D superlattice community.”

The main concern raised by reviewer #1 was that “...The main selling point of the paper, however, is an 18 nm pitch device. Unfortunately, the authors do not demonstrate a gate-gate map for this device, and several features of the magnetotransport are simply ignored. Mostly, the device resembles a normal graphene device with single or double alignment of the encapsulating hBN layers”.

The referee explained his concerns in further detail, saying that “Without gate-gate maps and additional characterization, there is no reason to believe that the features observed are not due to alignment with one or two of the encapsulating hBN layers. Furthermore, there are several features in the fan diagram that are completely ignored, e.g. several signatures of satellite fans between $n/n_0 = 0$ and 4”.

We understand the referee’s concern and agree it is critical to clearly differentiate our results from those possible to obtain with an hBN-aligned device. The revised manuscript conclusively shows the superlattice effects we see are induced by the electrostatic gate, based on new measurements and analysis.

Specifically, in the current manuscript we show the gate-gate dependence the referee asked for, which conclusively demonstrate the SL resistance peak only appears when voltage is applied to the patterned gate electrode. These new measurements are now incorporated in the main text (in Fig. 3f) and also replicated below, for the reviewer’s convenience (we note that displaying the results with multiple line traces is clearer in this case than with the gate-gate map the referee suggested, which is otherwise available now in the SI). Clearly, the evolution we see in the SL resistance peak as a

function of voltage applied to the patterned gate is not expected in a SL induced by hBN-alignment and identifies conclusively the role of the patterned gate in the measurement.

Figure 1 - Longitudinal resistivity vs. filling number for device 2, as a function of the patterned gate voltage, showing the evolution of the satellite peaks at $n = \pm 4n_0$ with increasing patterned gate voltage.

This is even more clear when we plot the excess resistivity, i.e. the resistivity difference at a certain PBG with respect to $PBG = 0$ V, as displayed below.

Figure 2 - Excess resistivity with respect to the $V_{PBG} = 0$ V case for Dev 2. Up to 100Ω excess resistivity is observed at $n/n_0 = \pm 4$, while it drops at higher carrier densities. Further experiments are needed to verify and reproduce the possible excess resistivity peak at $n/n_0 = \pm 2$.

Another important evidence that our measurements are not related to hBN-Gr alignment, comes from the charge density values. according to [Yankowitz, et al. " Nat. phys. 382 (2012)], the maximum Moiré length for a perfectly aligned graphene/hBN superlattice is 13.9 nm. Considering that graphene/hBN moiré superlattice is hexagonal, the ratio of the largest possible alignment superlattice to our 18 nm square superlattice is

$$\frac{A_{hBN/G}}{A_{Sq18}} = \frac{13.9^2 \sqrt{3}/2}{18^2} \approx 0.5.$$

Accordingly, the satellite peaks for a perfectly aligned graphene/hBN superlattice would appear at twice the density compared to the satellite peaks observed. If the lattices are not perfectly aligned, this ratio would be larger. The plot below shows a gate trace as a function of the carrier density and clearly shows the superlattice peaks we observe appear at carrier densities much lower than possible for alignment-induced superlattices. Importantly, the carrier density is conventionally extracted from hall effect measurements and a factor 2 error is not expected.

Figure 3 – Longitudinal resistivity vs. carrier density for device 2. Solid vertical lines indicate the density for 4 electrons (or holes) in a 18 nm square lattice, corresponding to $n/n_0 = \pm 4$ in Fig. 3 of the main manuscript (the location where Landau fans emerge from). Shaded purple region indicates where the satellite peaks for graphene/hBN superlattices can appear.

Furthermore, in the revised, manuscript we also include pictures of the specific flakes used for both heterostructures. The crystalline edge for the graphene flake composing device 2 is clearly identified and is seen to be misaligned with the encapsulating hBN flakes (see plot below, black dashed indicates graphene, red and green dashed lines indicate hBN). Below, we also show the shape of the graphene flake. We note that two edges of the flake were defined with an AFM tip (see picture) and are not natural edges of the graphene. This confirms that there is no alignment (single or double) of the graphene to the hBN.

Figure 4 – optical microscope pictures taken during the fabrication of device 2. Left is the full heterostructure, on top of the patterned gate. Scale bar is $5\mu\text{m}$. Black, green and red dashed

lines correspond to the crystalline edge of the graphene and the two hBN flakes, accordingly. Right is the graphene flake used. Black arrows indicate the location of cuts made with AFM tip.

In light of all the above evidence – the location of the density peaks, the influence of patterned gate voltage and the optical pictures of the flakes – we believe we can rule out the role of hBN alignment. However, we also acknowledge the reviewer was correct in raising this scenario and in the current manuscript we have included the essence of the arguments above.

In addition to this major concern, reviewer #1 has listed several additional issues, which in our understanding are of a more minor nature. Before responding to these in full detail, we just want to thank this reviewer for his/her unusually extensive report and the many useful comments he/she gave.

Reviewer #1 commented that: “9 of the 35 references are either review papers or textbooks, raising questions about how well the authors researched the literature”.

While we have made an extensive literature survey as part of this work, the literature on nanofabrication is, in all honestly, vast. The relatively large number of review papers and textbooks we cite reflects exactly that. It is difficult to name single papers which are the most relevant to a wide topic, e.g. to electron beam high resolution lithography. Nevertheless, we did add a few additional (non-review) references on nanofabrication, based on the referee’s suggestion.

Reviewer #1 wrote, regarding the abstract of our manuscript that our statement that on proximity effects typical to electron beam lithography is imprecise since “Suspended membranes also alleviate proximity effects in EBL”.

We were indeed comparing to conventional EBL using standard samples. We have modified the abstract accordingly.

Reviewer #1 commented:

In general, the introduction is littered with statements that are not backed up with any references.

“extreme ultraviolet optical lithography (EUV), the main lithography technique in the semiconductor industry.” DUV and EUV are not the same. EUV is used for only very few layers of the leading edge nodes at TSMC. Not even Intel is shipping products using EUV yet.

“This type of nanofabrication can achieve small feature sizes and is highly scalable, but is prohibitively complicated and expensive.”

What is the context here? EUV is expensive, indeed, but not prohibitively so for

industry. The introduction is quite messy, and mixes up the narrative between industry applications and academic prototyping needs.

The reviewer is absolutely correct that we have been referring to DUV, not EUV.

The purpose of comparison to STM or DUV lithography is to give proper background to the method we're suggesting. These are alternative methods that could at first glance be thought to be relevant, but are indeed not applicable to academic research\prototyping.

However, we do agree with the referee the introduction was mixing different narratives. We have revised it and hope the current version is clearer.

"However, two major limiting factors affect He-FIB-based nanofabrication: (1) Milling of materials directly on a substrate is accompanied by deposition of He bubbles inside the substrate, contaminating it and affecting surface topography. (2) Secondary collisions damage the quality and integrity of the patterned material. It is possible to reduce the damage by milling suspended materials"

At this point, only two papers are referenced about FIB lithography. Both extensively discuss suspended samples and membranes, overcoming (1) and (2). Why are these issues brought up?

There are three reasons to discuss suspended samples:

- (1) While it is technically possible to fabricate high quality suspended graphene/graphite and then directly mill them (e.g. the work of Mizuta, reference 17 of the current paper), making even a small-area suspended graphene device is a complicated process, especially if one wishes to keep the graphene clean (requiring current annealing and avoiding CVD graphene). Making these devices as a first step in a multi-part process seems to us to be an challenging, practically implausible, approach.
- (2) Moreover, direct milling of non-suspended samples is an obvious first approach for anyone attempting to use the high resolution of He-FIB in nanofabrication. In fact, we have tried this approach ourselves early in our experiments. We generally found that even if the graphite could be milled with very high precision, devices made based on directly patterned graphite showed a large degree of hysteresis, rendering them useless for measurements. This is, in our understanding, a result of damage incurred to the graphite by direct milling, which led to amorphization of the graphite and formation of charge traps. We included a brief account of our attempts in this regard in the SI and also added references from the literature
- (3) As explained in the revised manuscript, even suspended graphene incurs significant unintentional damage as it is milled. For example, even minute

amount of He ion can induce charge localization (see reference 25 of the revised manuscript).

Could you elaborate on the role of adding PMMA? I assume this serves as a sacrificial release layer?

The reviewer is exactly right about the purpose of the PMMA. This should have been clear from the text, but was omitted. We fixed this in the current version and thank the reviewer for noticing.

The patterned membrane is characterized by TEM, while the patterned PMMA/graphite stack is characterized by AFM. While AFM, through the FFTs, can to some degree give information about the geometrical disorder, it would be beneficial to do TEM imaging of graphite after pattern transfer, to measure the edge roughness and atomic disorder, as these are some of the main claims of the paper.

We thank the reviewer for his/her suggestion. We did try to image the patterned gate electrodes with TEM, but the process failed a few times during the lamella preparation step, destroying valuable samples in the process.

Also, given how pixelated the FFTs are, especially the inset, I don't think it is appropriate to use these for quantitative extraction of disorder. Zero padding the data prior to the transform might help here.

Regarding the FFT analysis, we agree with the reviewer that due to the limited AFM resolution/picture size restricts the quality of the FFT. We note we did use zero padding already in the original manuscript. However, we do believe the FFT does provides a quantitative measure of evaluating the degree of disorder in our system. The limited quality implies our results cannot determine the amount of disorder below a minimal amount (due to pixilation), but the FFT data still provides a valuable estimate of the minimal amount of disorder in the system, which is on the order of 3% period variation for the smallest (16nm) lattice.

Magnetotransport of device 1 looks convincing, especially with the gate-gate map in the supporting information, but the length scale has already been reported in the literature previously. The quality of the 47 nm device suggests that the presented method could give better device quality compared to the previously reported methods based on electron-beam lithography, but to support this, the authors would have to do a more careful analysis of both the transport data and the quality of the fabricated membranes and pattern transfers.

We thank the reviewer for his\her comment and we certainly agree that our method could lead to higher quality devices. Indeed, we think the quality of our superlattice is superior to previously reported ones [Huber et al. Nano Letters 2020]. However, some

variation in device quality (e.g. mobility) is usually expected, since parameters as hBN thickness or even the graphene/hBN sample quality plays an important role, making a direct comparison difficult. Likewise, detailed analysis of the quality of fabricated lattices is not always reported in EBL-based artificial SL, which greatly complicates the comparison.

Finally, the reviewer comments that: “there are several features in the fan diagram that are completely ignored, e.g. several signatures of satellite fans between $n/n_0 = 0$ and 4”.

We thank the reviewer for noticing and raising this point, which we also noticed and find quite interesting. We do not understand those small oscillations, but it is clear from the data that they appear only under magnetic field. We have discarded the possibility of a correlated state since the density at which they appear does not correspond to an integer number of electrons per SL unit cell. The magnetic field dependence seems to point towards some interference effect due to the carrier density modulation, i.e. periodic pn junctions [Young, A., Kim, P. Quantum interference and Klein tunnelling in graphene heterojunctions. *Nature Phys* **5**, 222–226 (2009)]. However, significant further research, both experimental and theoretical, is required to confirm the origin of this effect (or even whether it's not a measurement artefact of some sort). We therefore did not want to speculate in the manuscript on this effect, but we did add a comment in the main text noting on this phenomenon.

Reviewer #2 (Remarks to the Author):

In their manuscript "Engineering high quality graphene superlattices via ion milled ultra-thin etching masks", Ruiz et al present a fabrication method for extremely finely pitched 2D superlattices by using Helium (or Ga) FIB. By first milling an ultrathin membrane and only then transferring it to the target substrate, milling and removing the template, they circumvent the problem of extensive contamination or milling damage usually found in FIB based lithography. With their approach they can extend the range of achievable feature resolution from a ~ 35 nm lattice period, achieved in state-of-the-art e-beam lithography, down to about ~ 18 nm. They integrate their superlattices into a high quality graphene workflow and demonstrate successful definition of a tunable superlattice potential in graphene.

The fabrication and experimental results are astounding, and present a significant step forward in nanofabrication, at the expense of very specialised equipment (He-FIB) and a quite involved workflow. The results definitely merit publication, but I would like to suggest a few improvements, mainly regarding the presentation of the results.

We thank reviewer #2 for the very positive comments, and below we will try to address his suggested improvements for our manuscript

The reviewer states “In the Supplemental Material, the authors quite openly discuss a few issues with the presented technique (section 3, lines 80..) A summary of those (and a few more) points should make it into the conclusion of the main text, otherwise the conclusion sounds too optimistic. In addition to what is mentioned in the Supplemental Material, I would like to add that a 2D hole array is the most stable pattern imaginable. I think that fabricating, e.g., a stripe array with a similar pitch, would pose extreme challenges in terms of stability, and a non-connected geometry is evidently impossible to make with a stencil technique. The authors should address those issues in the main text, even though the new technique will still remain extremely useful for quite a few applications.”

We definitely agree with the reviewer on this point, and we have added a few lines in the conclusion into a new paragraph discussing the limitations and ways to improve our process. Indeed, this was not covered in our manuscript and we thank the reviewer for noticing it.

The reviewer proposes a way to improve our process “Can Si membranes be etched in the present workflow without damaging the substrate? This could remove the need for a PMMA based lift-off. Maybe some F-based dry etching could work?”

This is an interesting proposal, which we have actually tried ourselves before settling for the PMMA-based lift-off solution. As the referee predicts, silicon membranes can be etched with SF_6 . This can, in principle, lead to improved yields and resolution for our process. However, we found that the process required very precise monitoring in order to avoid excessively fluorinating the graphite surface, while etching the Si membrane completely. Hence, practically, we found the PMMA based option to be more reliable.

Here, the reviewer comments “Brown Zak oscillations (Fig 4, and lines 152...) are more easily observed at elevated temperatures, when Landau fans are already washed out. One can argue that BZ oscillations are just the remnants of the Hofstadter butterfly skeleton (which is what the authors actually see in Fig 4), but at least the authors of Ref 27 would probably disagree with this. Maybe some magnetic field sweeps at around 100 K and fixed density are available, or can be supplied?”

We thank the reviewer for this valuable observation. We are very interested in studying magnetotransport in such lattices, including at higher temperatures, as would be needed to study Brown Zak oscillations. However this work is still in the process and exceeds the scope of the current study. Hence we agree it is better to remove the statement on Brown Zak oscillations, until it can be demonstrated conclusively.

The reviewer requires more detail about a variation of the fabrication technique “The authors briefly mention that they also tried Ga FIB (which is available in more labs than He FIB). Can they be more specific as to the minimum feature size realized here, and supply some images? The present statement in SM, l. 52 is too vague.”

In our experimental work we have been able to generate periodic lattices as small as 45-50nm. However, this is almost certainly due to limitations of the specific Ga FIB used for this work (e.g. alignment issues with low current apertures). We expanded our explanation in the SI to clarify better the limitation of Ga FIB.

Finally, reviewer #2 asks to add the following information to the manuscript “Can the authors provide the thicknesses of hBN separating patterned gate and graphene channel? They can be inferred from I. 132 and I. 133 but the authors could just as well provide the values.” And to provide the following “Also, real images of the actual devices should be supplied, at least in the Supplemental Material.”

We thank the reviewer for these comments, and we have incorporated both. We specify the hBN thickness in the main text, and include device pictures in the supplementary material.

Reviewer #3 (Remarks to the Author):

The reviewer #3 supports the publication of our results in Nature Communications “In this paper, a new method of fabricating superlattice devices is reported, in which a suspended Si membrane is etched by ion milling to create an etching mask. This mask is transferred onto a thin graphite flake so that it can be used to etch a superlattice gating structure. Using this technique, the authors are able to create a lithographically defined superlattice device with significantly shorter periodicity than other methods with similarly low disorder, namely methods relying on e-beam lithography to define the mask.

The measured device resistance (Figure 4) indicates a Hofstadter butterfly spectrum as is expected in these devices. Given the short periodicity in device 2 (18nm), which increases the spacing of fractal band features in both electron density and magnetic field, it is rather disappointing that the Hofstadter effect is so weak. I would hope that the smaller periodicity would give rise to a more brilliant butterfly; that this increased spacing would allow for greater resolution of fractal subbands. The authors give a well reasoned explanation for why this effect weakens as the periodicity is shortened. This perhaps elucidates an inherent weakness of using electrostatically induced SLs as a means of replicating the effects of small-angle moire superlattices.

Regardless, this fabrication technique is a significant step forward for patterning gating structures with nanometer scale features. There is a wide audience in the layered heterostructure devices community who would be interested in this result and potentially find use for this technique, especially since it can be applied to more than superlattices. Therefore, I support its publication in Nature Communications.”

We deeply appreciate the reviewer’s comments and we are happy to hear that the one of the pioneers of graphene band structure engineering found our technique to be of interest. As the referee stated, shortening the period is indeed not the path forward to observe more brilliant butterflies. However, we do wish to stress that our work paves

the way to observe Hofstadter butterflies (and related effects) in non-trivial lattices (e.g. honeycomb, Lieb), as well as in gapped materials.

Below, we aim to address the very detailed comments of the reviewer:

The reviewer wrote “39- Opening sentence states a trade-off, resolution vs throughput. EUV is given as an example of the low-resolution, high-throughput end of the spectrum, but it's described as having “small feature sizes” but complicated and expensive. Perhaps give actual numbers for the features sizes STM vs EUV to elucidate this comparison?”

More generally, I found this discussion of resolution vs throughput somewhat irrelevant to the academic setting where EBL is very common. I never considered using STM for making a superlattice (should I?)”

We fully agree and have modified that part accordingly. We do not discuss EUV anymore, but UVL. We just introduce STM and UVL as examples of alternative methods nanofabrication. Indeed, STM is not practically applicable for superlattice research, due to its extremely low yield. We do however think it is important to mention these alternative methods, to give proper context to our findings.

The reviewer points “71- Specifically, how does the transport data indicate minor damage to the electrodes? Can you measure the mobility in your devices (or mean free path)?”

From your transport data, it is clear that there is no breaking of graphene's 4-fold degeneracy, which sets these devices apart from moire superlattice devices. Of course, this is true of all previous attempts to create a lithographically defined superlattice as well. My point is, there is still a huge gulf between a normal encapsulated graphene device (with top and bottom graphite gates, no less) and devices like these, in terms of mobility, degeneracy breaking, and true quantization of Hall resistivity. Pointing to specific signifiers of device quality is valuable here.”

We thank the reviewer for his comments. We understand that a signifier of the quality is the observation of Landau fans emerging from satellite peaks. The fact we can observe them tells us about the an upper of disorder induced in the superlattice energy bands. Evaluating mobility with standard methods would result in low values when the patterned gate is applied. That is also the case in moiré superlattices, where scattering processes lower mobility [see *Wallbank, J.R., et al. Nature Phys 15 (2019)*]. In the absence of patterned gate, we find mobility on the order of $60,000\text{cm}^2/\text{Vs}$, or higher for both devices (with the patterned gate voltage turned off), but this appears to depend more on the quality of the stack. For example, new patterned gate devices made for new projects (but using the same fabrication methods) can achieve much higher mobility, essentially limited by device size.

The reviewer asks to specify the following in the text “90- Can you just state the thickness of the hard mask here? I think it's helpful context for the reader. Looking at the supplementary, it's of order 10 nm?”

We have now added the thickness explicitly in the main text.

The reviewer noticed in our text “119- “On the other hand, the PBG only changes the carrier density on the areas of graphene right on top of the remaining graphite, while the areas on top of the patterned holes remain unaffected”

This isn't really true though, right? Looking at your simulation results (Figure S3), you can see that the carrier density within the holes is affected by the graphite gate voltage as well. I think if your model was 3D with an actual circular hole, I think you'd see an even larger effect within the hole region.

It's fair to say that you can independently tune the carrier density and SL strength, but the effect of the PBG is more complicated than it only affecting the graphene above the unetched regions. This thickness, t_{BN} , is also critical here for how much the hole regions are affected by PBG voltage.”

This sentence was indeed not fully accurate: what we meant was that with both gates we can tune independently carrier density and SL strength. We have changed the text accordingly.

The reviewer wrote about the disorder “Fig 1, line 70, line 80 - This paper's characterization of disorder in these samples is lacking any mention of the disorder created by RIE etching of the PBG or exposing it to PMMA. I am not sure where in the paper this could be added, but I think a recognition of this source of disorder would be valuable. The text mentions the importance of considering charge traps (line 70) and external contamination (line 80), both of which I think are potentially created through the PMMA and RIE etch. Though I recognize that annealing the PBG following the acetone liftoff is fairly effective, it is still worse than no PMMA contamination. Also, this sort of contamination is shared by the EBL defined techniques which are referenced in this paper.

To be clear, this technique is still very valuable, as it allows FIB to be used in devices with similar quality to those produced by EBL; so Figure 1 is still valid.”

We thank the reviewer for his comments. In fact, the disorder induced by the RIE process is shared with EBL, even though we can use thinner etching masks since silicon is resistant to O_2/Ar . This means we partially circumvent aspect ratio issues, but we still suffer from RIE ions scattered by the SiO_2 . We have added a paragraph to the conclusion, stating these issues and possible ways to solve them.

The reviewer suggests how to improve a schematic “Fig. 3a: This attempt to show the device, using transparent layers, is not really legible. I don't think it will be helpful for anyone who doesn't already know how these devices are stacked. Perhaps try

removing metal leads and the “Hall bar” shape of the device, and just focus on the layers themselves.”

This is an excellent suggestion, which we have implemented. We fully agree it improves the figure substantially.

Also, the reviewer noticed a label which is not visible enough “to Fig 4a- The red ϕ/ϕ_0 labels are unreadable. Make them bigger? Bold font? Make them another color (white?)?”

This is now corrected, we thank the reviewer for noticing this issue.

Finally, the reviewer gave a positive feedback about our conclusion and outlook “I like the Conclusion of the paper. I think that looking at new lattice geometries and Fermi-Hubbard model simulators are promising next steps for this sort of FIB mask transfer technique.

I also think that this paper shows a difficult hurdle for engineering Hofstadter butterflies: that the lattice strength at the cloned Dirac bands weakens as the SL period is shortened. This is all the more reason to shift attention to new applications for this technique.”

We could not agree more with the reviewer on this point, and we are happy to share the same vision on towards where the field should develop.

List of changes

Track changes was activated in the document. Changes are listed here in addition, for convenience.

Main text (numbers indicates lines involved)

- Section headings renamed according to Nature Communications style.
- 15. Added “standard” to electron beam lithography (suggested by #1).
- 18-19. Re-written sentence regarding the concerns of reviewer #1 about the quality of device 2.
- 43. “extreme” removed, now we speak about ultraviolet lithography (suggested by #1).
- 44-46. Sentence removed, since EUV was not very relevant in this context. (suggested by #1 & #3).
- 92. Silicon membrane thickness is now indicated (suggested by #3).
- Purpose of the PMMA layer is clearly specified (suggested by #1) and added note about etching alternatives

- 111-1112. hBN thickness written explicitly (suggested by #2).
- Removed claim of Brown Zak oscillations as we do not have high temperature data supporting the claim (suggested by #2).
- 170-176. Discussion on limitations of the technique due to the suspended nature of the masks, sources of disorder and contamination, and possible routes to solve it. (suggested by #2 & #3)

Figures

- Figure 1. Format updated for 1 column width.
- Figure 2. Format changed and inset in panel f removed.
- Figure 3a. Schematic has been modified according to reviewer #3 suggestion.
- Figure 3f. has been replaced with a gate dependence multi-trace, following suggestions of reviewer #1
- Figure 3 panels updated to comply with Nature communications format.
- Figure 4. Font sizes adjusted to Nature Communications style. Red labels on the left panels are now more visible (suggested by #3).

Supplementary

- Added section 4 with optical images of the heterostructures before and after defining the hall bars (suggested by #2).
- Discussion on Ga FIB milling alternative.
- Added brief discussion on etching alternative to PMMA liftoff.
- Added hall measurements (previously Fig. 3f of main text).

Reviewers' Comments:

Reviewer #1:

Remarks to the Author:

I thank the authors for their rebuttal and for addressing my concerns. I am mostly satisfied with the rebuttal and have just a few comments below.

Data:

The additional data for the 18 nm device, in terms of resistivity vs. density for different PBG voltages, is indeed more convincing and helps to demonstrate the presence of a superlattice that is gate-tunable. However, the quality of the gate-gate map for Dev 2 is very concerning, especially considering the high quality for Dev 1. Since the main novelty of the paper is the low length scale of Dev 2, I believe the paper would be significantly stronger if the authors had the possibility to obtain or redo the Dev 2 measurements. However, I acknowledge that extra measurements are not likely to be added to the manuscript.

The authors argue that there is no alignment to hBN, but I will just briefly note that it is not uncommon to have moiré superlattices larger than 13.9 nm, as the presence of hBN on both sides of the graphene can lead to a super-superlattice. I'm not sure it is relevant here, but I just wanted to add it as a note for future work.

Furthermore, care should be taken when relying on the edges of hBN and graphene to avoid or obtain alignment. While graphene often tears along armchair and zigzag crystal directions, it is also very common to find edges that are 15° relative to each other, which would be a perfect mix of armchair and zigzag. The 72° (aka 12°) alignment would, in that case, turn into a 3° alignment - given uncertainties, this is small enough to argue for the presence of a moiré superlattice.

Having said all that, I believe the authors have done their due diligence in avoiding alignment, albeit I am not 100% convinced that there might not also be some moiré superlattice effect on top of the artificial superlattice effects. This concern is mostly from the gate-gate map, where, to my eyes, it looks like there are additional satellite peaks that trace to 0V on the PBG, indicating that they are not gate-tunable, which in turn means they are of moiré origin. Higher-quality data, or an additional device, would remove these concerns for me.

References:

I acknowledge the vastness of the literature and am not against a few review papers and/or textbooks, but the use was (is?) excessive in the current manuscript.

Reviewer #2:

Remarks to the Author:

The authors have addressed the concerns by all referees in great detail and give convincing answers. I am fully satisfied with their response and recommend the article for publication.

Reviewer #3:

Remarks to the Author:

I am satisfied with the changes and additions made to the manuscript in response to my previous comments. The discussion of alternate fabrication methods reads much better now. I also appreciate the expanded conclusion, which now more candidly states the limitations and future

challenges of this fabrication method. This is a clear benefit to readers wanting to emulate and improve on this work.

I also appreciate the new measurements showing the formation of satellite peaks with PBG voltage (Figure 3f).

I maintain my support for publication.

Reviewer #1 (Remarks to the Author):

“I thank the authors for their rebuttal and for addressing my concerns. I am mostly satisfied with the rebuttal and have just a few comments below.”

We thank the reviewer for his/her comments that really helped to improve the quality of the manuscript (especially the introduction and Fig. 3f). We are very happy that she/he is now satisfied with the current version of the manuscript.

The reviewer states “The additional data for the 18 nm device, in terms of resistivity vs. density for different PBG voltages, is indeed more convincing and helps to demonstrate the presence of a superlattice that is gate-tunable. However, the quality of the gate-gate map for Dev 2 is very concerning, especially considering the high quality for Dev 1. Since the main novelty of the paper is the low length scale of Dev 2, I believe the paper would be significantly stronger if the authors had the possibility to obtain or redo the Dev 2 measurements. However, I acknowledge that extra measurements are not likely to be added to the manuscript.”

We agree that in the first version the data for different PBG was missing, and once included the manuscript is more convincing to the reader. We are working in different directions to improve our technique.

The reviewer comments about the possibility of a double alignment to the hBN flakes “The authors argue that there is no alignment to hBN, but I will just briefly note that it is not uncommon to have moiré superlattices larger than 13.9 nm, as the presence of hBN on both sides of the graphene can lead to a super-superlattice. I'm not sure it is relevant here, but I just wanted to add it as a note for future work. Furthermore, care should be taken when relying on the edges of hBN and graphene to avoid or obtain alignment. While graphene often tears along armchair and zigzag crystal directions, it is also very common to find edges that are 15° relative to each other, which would be a perfect mix of armchair and zigzag. The 72° (aka 12°) alignment would, in that case, turn into a 3° alignment - given uncertainties, this is small enough to argue for the presence of a moiré superlattice.”

We agree that there is in principle a possibility that the system would be aligned to both flakes, despite evidence to the contrary (optical microscope pictures and transport measurement). While find it unlikely, we are now discussing this possibility in main text (l. 129-133).

As the reviewer states “Having said all that, I believe the authors have done their due diligence in avoiding alignment, albeit I am not 100% convinced that there might not also be some moiré superlattice effect on top of the artificial superlattice effects. This concern is mostly from the gate-gate map, where, to my eyes, it looks like there are additional satellite peaks that trace to 0V on the PBG, indicating that they are not gate-tunable, which in turn means they are of moiré origin. Higher-quality data, or an additional device, would remove these concerns for me.”

Even if that is the case and there is a double-alignment, it would be on top of the artificial superlattice effects, which are conclusively demonstrated through the PBG dependence in Figure 3f, Supplementary Figures 10 and 11.

Reviewer #2 (Remarks to the Author):

The reviewer is fully satisfied with the changes and still recommends its publication “The authors have addressed the concerns by all referees in great detail and give convincing answers. I am fully satisfied with their response and recommend the article for publication.” We thank again reviewer #2 for the useful advice that helped improving the first version of the manuscript.

Reviewer #3 (Remarks to the Author):

Reviewer #3 states “I am satisfied with the changes and additions made to the manuscript in response to my previous comments. The discussion of alternate fabrication methods reads much better now. I also appreciate the expanded conclusion, which now more candidly states the limitations and future challenges of this fabrication method. This is a clear benefit to readers wanting to emulate and improve on this work.” We completely agree that all those changes improved the quality and readability of the manuscript, and we thank him for his advice that contributed to that improvement.

Reviewer #3 also maintain his support for publication “I maintain my support for publication.”